# K-RBBSO Algorithm: A Result-Based Stochastic Search Algorithm in Big Data

**Sungjin Park** 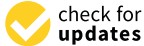 **and Sangkyun Kim** *

Graduate School of Business, Kyung Hee University, Seoul 02447, Republic of Korea
* Correspondence: saviour@khu.ac.kr; Tel.: +82-02-961-0127

**Abstract:** Clustering is widely used in client-facing businesses to categorize their customer base and deliver personalized services. This study proposes an algorithm to stochastically search for an optimum solution based on the outcomes of a data clustering process. Fundamentally, the aforementioned goal is achieved using a result-based stochastic search algorithm. Hence, shortcomings of existing stochastic search algorithms are identified, and the k-means-initiated rapid biogeography-based silhouette optimization (K-RBBSO) algorithm is proposed to overcome them. The proposed algorithm is validated by creating a data clustering engine and comparing the performance of the K-RBBSO algorithm with those of currently used stochastic search techniques, such as simulated annealing and artificial bee colony, on a validation dataset. The results indicate that K-RBBSO is more effective with larger volumes of data compared to the other algorithms. Finally, we describe some prospective beneficial uses of a data clustering algorithm in unsupervised learning based on the findings of this study.

**Keywords:** data clustering; unsupervised learning; big data; biogeography-based optimization; silhouette coefficient

## 1. Introduction

Clustering is an unsupervised learning algorithm that evaluates the representativeness of data and classifies them on this basis [1]. Many researchers have approached metaheuristics-based clustering to solve complex problems. According to Jahwar and Abdulazeez [2], the metaheuristic method for clustering is a concept that is the basic framework of an algorithm designed and structured to solve problems by organically acting on various complex problems; that is, it is not limited to specific problem-solving. The reason for using metaheuristics algorithms is to determine the best solution and resource optimization for their effective use, reducing time consumption [3,4].

According to Hussain et al. [5], the domains of problem-solving using the metaheuristic algorithm are approximately 14 in number, and the biology domain accounts for the largest portion. Moreover, the basic model for designing a new metaheuristics algorithm is derived with 10 domains. The largest portion of new metaheuristic algorithm design models is ordered by insects, natural evolution, animals, and birds. Furthermore, clustering algorithms are divided into two main types: hierarchical and partitional clustering algorithms. These two types are currently the most used clustering algorithms. Hierarchical clustering is divided into agglomerative and divisive clustering. Partitional clustering is divided into hard/crisp, mixture resolving, and fuzzy clustering [6], for example, particle swarm optimization [7], the firefly algorithm [8], and group search optimization [9], which are all nature-inspired algorithms. The particle swarm optimization algorithm is developed to simulate and check the behavior of humans. The firefly algorithm mimics the characteristics of bright fireflies and their habit of gathering in bright light. Group search optimization models the hunting phenomenon of animals in groups.

In recent years, the clustering algorithm has been extensively employed by customer-facing businesses and organizations who wish to appropriately classify customers to deliver

personalized services. Current categorization approaches categorize and label objects based on predetermined criteria. As a result, the underlying information is not up-to-date and suffers from human bias. To resolve these problems, clustering is frequently utilized to reduce human cognitive mistakes and deliver personalized goods and services [10].

Several factors affect clustering quality (e.g., the number of clusters and the evaluation value). The number of clusters may be changed from a minimum of two to a maximum of n by the user. Clusters are formed by examining data features based on the number of clusters specified by the user. The evaluation value is a variable that determines whether the user-specified number of clusters is used. According to Hruschka et al. [11], it is reasonable for the time required for clustering to increase with an increase in the quantity of data. However, in the case of three or more clusters, the number of iterations required for optimal solution search increases exponentially with the quantity of data. A solution to this problem is expected but not definite.

Several studies have attempted to reduce the execution time of clustering. Optimization algorithms have been proposed for this purpose. Further, the result evaluation function has been analyzed to ensure clustering reliability. The Euclidean distance between two points on the coordinate plane is the most commonly used evaluation function. However, it has not been extensively used because the evaluation function diminishes with an increase in the clustering verification functions. Instead, the silhouette coefficient has often been employed, which takes values between −1 and 1 based on the relative distance between the data and the cluster [12].

The following issues have been reported in methods proposed to improve the clustering efficiency:

- Problem 1: The silhouette coefficient is not suitable for application to the evaluation value-based stochastic optimum solution search algorithm.
- Problem 2: As the evaluation of the silhouette coefficient requires significant computation, high computation time is required when applied to big data.
- Problem 3: Stochastic search for an optimum solution corresponding to a random initial solution is very time-consuming.

Problem 1 is related to the silhouette coefficient and the stochastic search equation of the optimization algorithm. In general, the optimization algorithm performs a stochastic search based on the initial search result. Most evaluation functions search for the optimal evaluation value by progressing from high to low values. For example, when the Euclidean distance is used as the evaluation function for data with k clusters, if the first and second clustering evaluation values are 80 and 70, respectively, the second clustering is considered to be superior to the first one for k clusters. However, the case of the silhouette coefficient is different—its search begins at −1 and progresses toward 1. At this point, the algorithm used to determine the probability of identifying the solution in the next iteration based on the result relies on the accepted evaluation value. By design, the evaluation value progressively decreases in a general stochastic search model. Conversely, the evaluation value of the silhouette coefficient increases over time. As a result, the existing probability search formula cannot be used in this case. Moreover, when the silhouette coefficient is applied to the search probability equation based on the evaluation value, a comparison of the random probability in (0, 1) with the search probability yields a value that is unconditionally larger than the former. Consequently, the algorithm does not function properly.

Two representative algorithms are associated with Problem 1—simulated annealing (SA) and artificial bee colony (ABC). SA [13] mimics the manufacture of steel products. By design, the algorithm identifies stochastically better solutions by accepting an evaluation value that is stochastically worse than the current evaluation value based on Equation (1):

$$e^{-(f(n)-f(e))/T},\qquad(1)$$

where f(n) denotes the previous evaluation value, f(e) denotes the current evaluation value, and T denotes the current temperature. In the SA algorithm, if the value of the

expression in Equation (1) is smaller than the randomly produced value between 0 and 1, the stochastically poorer evaluation value is adopted; otherwise, the current evaluation value is used until the termination condition is met. Next, let us consider the application of the silhouette coefficient to this equation. Assume that the previous evaluation value is −0.7, the current evaluation value is −0.1, and T = 0.8. Then, Equation (1) exhibits a value exceeding 1, and thus, the worse solution is unconditionally accepted. As a result, the corresponding formula must be modified to make it suitable for the application of the silhouette coefficient.

The ABC [14] algorithm mimics the honey-gathering process of honeybees. It identifies optimal solutions stochastically. By design, several on-looker bees (OBs) may be selected during the solution search, and one of them is stochastically searched for. The following probability equation is used for the stochastic search:

$$P_i = \frac{f(i)}{\sum_{j=1}^{OB} f(j)},\tag{2}$$

where P denotes the search probability for each OB and f(i) denotes the evaluation value corresponding to the solution identified using the $i^{th}$ OB. The denominator is defined to be the sum of the evaluation values corresponding to the solutions identified based on all previous OBs. A stochastic search is performed using this probability. However, the silhouette coefficient cannot be applied without modifying the formula. Further, as in the case of the SA algorithm, when the silhouette coefficient value exceeds a specific threshold, the solution obtained based on other OBs is no longer searched—only that obtained based on a single OB is searched repeatedly.

Problem 2 concerns a shortcoming of the silhouette coefficient itself. In the algorithm devised by Rousseeuw [12], the silhouette coefficient between each data point x in the data set S(x) and each other data point must be determined. Consider the clusters A, B, and C, where a(x) denotes the average distance between the data point x belonging to cluster A and other data points in the same cluster and d(x, B) and d(x, C) denote the average distances of x and each data point in clusters B and C, respectively. In this case, if d(x, B) < d(x, C), then b(x) = d(x, B). When these conditions are satisfied, the silhouette coefficient corresponding to the data point x may be calculated using Equation (3), and the final silhouette coefficient for all data can be calculated using Equation (4).

$$S(x) = \frac{\{b(x) - a(x)\}}{max\{a(x),\ b(x)\}}\tag{3}$$

$$Maximize\ \left(\frac{1}{n}\right) \sum_{i=1}^{n} S(x_i)\tag{4}$$

However, as stated above, because the computation of the silhouette coefficient involves the computation of the average distance between every data point in the dataset and the data point x, the computation time increases exponentially with the size of the data set. Further, the optimum solution cannot be identified until shortly before the completion of the computation based on the termination condition of the clustering algorithm to which the silhouette coefficient is applied. Moreover, the algorithm that performs a search based on previous results records many instances of literation on its own. As the number of iterations increases, the calculation time is increased [15]. Therefore, if the silhouette is applied, a high calculation time is required. The problem of increasing the calculation time because of the silhouette and the amount of data should be considered [16].

Problem 3 pertains to the cluster development procedure. Typically, an initial solution is determined using a random function prior to clustering. In the case involving three clusters, the solution is expressed in the form "1, 2, 3" or "0 0 1" to indicate the cluster to which each data point belongs. The difficulty lies in the determination of the initial solution.

The probability of discovering an optimal solution is high when the initial solution is good; otherwise, the process becomes very time-consuming.

In this study, we introduce the k-means-initiated rapid biogeography-based silhouette optimization (K-RBBSO) algorithm to resolve the three aforementioned problems. First, Problems 2 and 3 are solved by identifying the initial value using a single iteration of k-means. The k-means algorithms perform clustering based on the average value of the data belonging to each cluster [17,18]. In this case, the computation required to evaluate the silhouette coefficient is reduced by controlling the randomness of the initial solution generation. After identifying the initial solution, the computation speed is improved using the migration and mutation rules of biogeography-based optimization (BBO) introduced by Pal & Saraswat [19]. Subsequently, the evaluation function is changed from the existing Euclidean distance to the silhouette coefficient. BBO easily identifies an optimal solution when the number of clusters required for clustering is known. However, a fixed number of clusters is not applicable in our case—this must be obtained based on experiments.

## 2. Methods: K-Means-Initiated Rapid Biogeography-Based Silhouette Optimization

According to Pal and Saraswat [19], the habitat, which signifies solution expression, searches for better solutions via migration and mutation, and changes or adapts solutions with poor evaluation values into those with good evaluation values. Habitats with good evaluation values preserve their existing states by reducing the immigration rate $\lambda$ relative to other habitats and increasing the emigration rate $\mu$ relative to habitats with different solutions, thereby adjusting the solution values corresponding to other habitats and improving their evaluation values.

BBO changes and adapts the solutions by adjusting the immigration and emigration rates of habitats of solutions with good and poor evaluation values. This gives it an advantage over conventional algorithms, which search for new solutions completely randomly. In this study, the algorithm was modified to enable the application of the silhouette coefficient to the immigration, emigration, and mutation rates introduced by Pal & Saraswat [19] and Simon [20].

The procedure of the algorithm devised in this study is as follows.

- Step 1. Configure parameters such as the initial solution and the termination condition.

The parameters required to operate the algorithm are configured. These include the number of data points to be analyzed (row); the number of attributes (col); the number of clusters K; the number of habitats to be used in the K-RBBSO algorithm execution stage H; the initial solution for each habitat; and the termination condition.

Any desired condition can be selected as the termination condition. The initial solution is selected randomly. As described above, if K = 3, the solution expression matrix for row x K is denoted by (1, 2, 3, . . . , n) or (010, 100, 001).

- Step 2. Execute k-means to determine the initial solution.

K-means is executed for $H_1 \sim H_n$ for the habitat set H defined in Step 1. Here, the termination condition of k-means is taken to be "A = B if B > A" when the current evaluation value A is compared with the new evaluation value B. Moreover, if A does not depend on the row number, K-means is terminated, and the final solution is selected as the initial solution for the habitat. After the initial solution for each habitat is selected, the silhouette coefficient for the selected initial solution is calculated using Equations (1) and (2).

- Step 3. Select the emigration habitat.

Once the initial solution for the initially established habitat H has been established, the emigration habitat is selected. First, the silhouette coefficients of $H_1 \sim H_n$ are determined for the habitat H for which the initial solution has been established using Equations (1) and (2). The silhouette coefficient for each habitat H is taken to be $S_{H_n} = \text{Silhouette}(H_n) + 1$. This is because the habitat search process does not function properly if the silhouette coefficient becomes negative.

First, the emigration habitat selection probability is determined using the following equation:

$$E_{H_n} = \frac{S_{H_n}}{\sum_{j=1}^{n} S_{H_n}} \tag{5}$$

The probabilities of $E_{H_1} \sim E_{H_n}$ are determined using Equation (5), and one migration habitat is selected by comparing the probabilities with that of a random function in (0, 1).

- Step 4. Select the immigration habitat.

The silhouette coefficients of initial habitats $H_1 \sim H_n$ are selected before selecting the emigration habitat using Equations (1) and (2). Then, the immigration habitat selection probabilities $I_{H_1} \sim I_{H_n}$ are determined based on the silhouette coefficient for each habitat using the following equation:

$$I_{H_n} = \frac{1/S_{H_n}}{\sum_{j=1}^{n} 1/S_{H_n}} \tag{6}$$

Once $I_{H_1} \sim I_{H_n}$ are determined using Equation (6), one immigration habitat is selected by comparing the probabilities with those of a random function on (0, 1).

- Step 5. Perform migration for the selected habitat by selecting the center point.

Migration is performed for the emigration and immigration habitats selected in Steps 3 and 4, respectively. First, the average points (no medoid) of the emigration and immigration habitats are calculated. Then, one average point from each habitat is changed. Subsequently, the changed average point is added to the immigration habitat, and its silhouette coefficient is calculated. If this improves the evaluation value, the current immigration habitat solution is updated.

- Step 6. Select the mutation habitat.

Following the migration of the emigration and immigration habitats, the mutation is performed by selecting the mutation habitat. For the stochastic selection of the mutation habitat, $M_{H_1} \sim M_{H_n}$ are determined using the following equation:

$$M_{H_n} = \frac{1/S_{H_n}}{\sum_{j=1}^{n} 1/S_{H_n}} \tag{7}$$

The mutation habitat is selected by comparing $M_{H_1} \sim M_{H_n}$ determined using Equation (7) with the probability of the random function (0, 1). Then, the mutation is performed for the selected mutation habitat.

- Step 7. Perform mutation.

The silhouette coefficient corresponding to each data point is determined using Equation (3) around the habitat solution selected in Step 5. Then, the data points with the lowest silhouette coefficients are replaced with randomly selected data points belonging to other clusters. Subsequently, the silhouette coefficient is calculated again using Equation (3). If the evaluation value has not improved, the previous data point is replaced with another data point; otherwise, the altered solution is updated to the mutation habitat.

- Step 8. Regenerate the habitat.

The silhouette coefficient for each habitat is calculated using Equations (3) and (4). For the habitat with the worst evaluation value, the previously stored solution is initialized, and the initial solution of the step is selected again using the random function.

- Step 9. Select the final solution.

At this stage, the output comprises the evaluation value and solution corresponding to the generated habitat with the highest evaluation value. Now, the fulfillment of the termination condition defined in Step 1 is verified, based on which the algorithm is either continued or terminated. If it is continued, Steps 2–8 are repeated to identify a better

evaluation value. If it is terminated, the best evaluation value and solution identified by the algorithm are presented as the output, and the procedure is terminated.

## 3. Results

To verify the performance of K-RBBSO, its stochastic search algorithm based on the validated data and the current evaluation value were compared with those of RBBSO without k-means. Windows 10, 64-bit processor, Intel® i5-1240P CPU, 1.7 GHz, and 16 GB RAM were used as part of the experimental environment. The cluster engine was developed using C++ 2022 (64-bit) version 17.3.5.

As verification data, the dataset of the widely used UCI machine learning repository was employed (Figure 1). It includes Iris, Wine, Glass, Vowel, Cloud, and CMC, which have been utilized in data clustering research. The data used in this study are summarized in Table 1.

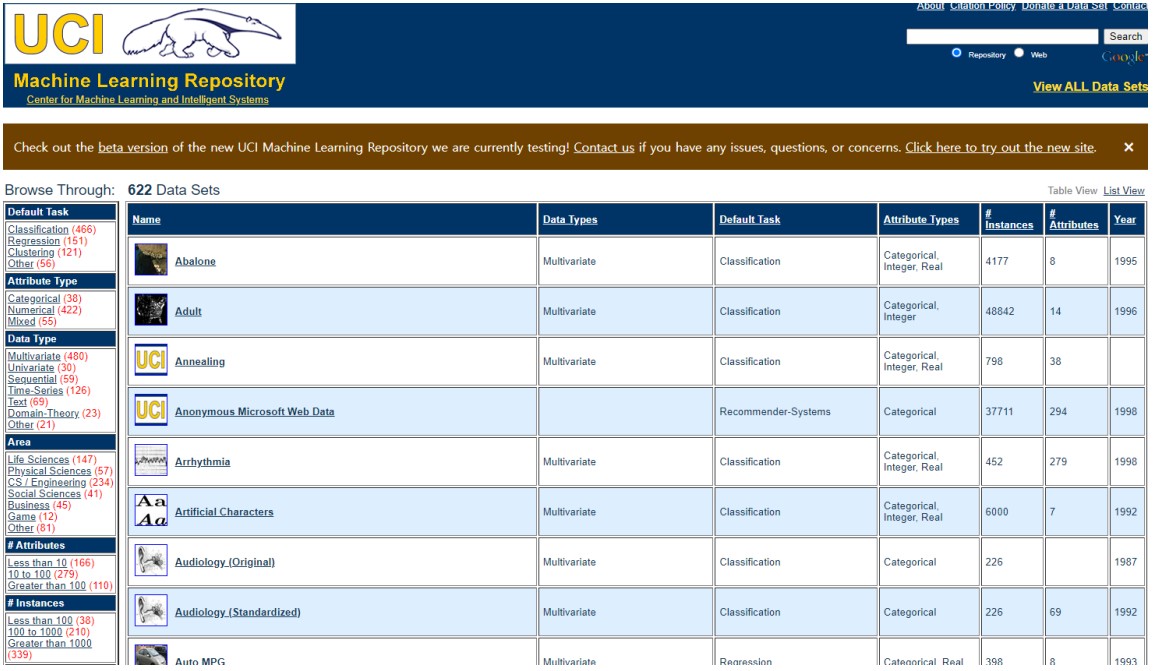

**Figure 1.** UCI Machine Learning Repository Home Page.

**Table 1.** Data and algorithm parameter description.

| Data Name | # of Data | # of Attributes | # of K | Reference |
|---|---|---|---|---|
| Iris | 150 | 4 | 3 | |
| Wine | 178 | 13 | 4 | |
| Glass | 214 | 9 | 6 | |
| Vowel | 871 | 3 | 6 | [21] |
| Cloud | 1024 | 10 | 10 | |
| CMC | 1473 | 9 | 3 | |

| Algorithm Name | Parameter 1 | Parameter 2 | Parameter 3 | Parameter 4 |
|---|---|---|---|---|
| K-RBBSO | # of habitats = 10 | # of emigration habitats of each calculation = 1 (worst) | # of immigration habitats of each calculation = 1 (best) | # of mutation habitat of each calculation = 1 (best) |
| RBBSO | | | | |
| S.A. | T = 1 | $\triangle$t= 0.99 | t = 100 | |
| A.B.C. | # of pop = 5 | # of bob = 5 | # of iterations limit of each pop = 100 | |

The number of habitats of the K-RBBSO and RBBSO algorithms was set to 10. In addition, the number of emigration, immigration, and mutation habitat was set to 1. Each habitat choice function was applied according to steps 5, 6, and 7. The large t of the S.A. algorithm was set to 1, and the delta t was set to 0.99. The delta t in the S.A. algorithm represented the falling temperature in every calculation. The small t represented the calculation iteration duration for a calculation time of delta t. In the A.B.C. algorithm, the numbers of pop and bob were set to 5. Moreover, each iteration limit was set to 100. Every parameter between S.A. and A.B.C. was a measurement obtained from the empirical experiment in this study.

To ensure validity and reliability during the data selection process, various numbers of data points and attributes were considered. Moreover, as the aim was the verification of effectiveness, the number of clusters K was established for each data type by referring to previous research. For instance, if a previous study on Iris data had used three clusters, for instance, K was taken to be 3 in this study.

Using the aforementioned data, the performances of SA [13], ABC [14], and RBBSO without K-means algorithms were compared with that of the proposed algorithm. Additionally, the termination condition for all algorithms was set to "terminate when the same result value is obtained corresponding to the number of rows of the presently running data". For instance, the algorithm was programmed to terminate if the evaluation value of the execution result was produced 1473 times during the analysis of CMC data.

In addition, the silhouette coefficient and computation times were concurrently monitored and compared for effectiveness analysis. Each algorithm was executed ten times for each data type, and the mean, standard deviation, and minimum and maximum values were recorded. The outcomes of each analysis are presented in Tables 1 and 2, respectively.

**Table 2.** Estimated values of the silhouette coefficient.

| Data Name | | K-RBBSO | RBBSO | SA | ABC |
|---|---|---|---|---|---|
| Iris | Average | 0.553 | 0.553 | 0.627 | 0.178 |
| | S.D. | 0.000 | 0.000 | 0.000 | 0.260 |
| | Min | 0.553 | 0.553 | 0.627 | −0.020 |
| | Max | 0.553 | 0.553 | 0.627 | 0.586 |
| Wine | Average | 0.564 | 0.564 | 0.582 | −0.009 |
| | S.D. | 0.002 | 0.002 | 0.000 | 0.028 |
| | Min | 0.562 | 0.562 | 0.582 | −0.031 |
| | Max | 0.567 | 0.567 | 0.582 | 0.059 |
| Glass | Average | 0.556 | 0.553 | 0.027 | −0.036 |
| | S.D. | 0.051 | 0.059 | 0.000 | 0.002 |
| | Min | 0.479 | 0.450 | 0.027 | −0.039 |
| | Max | 0.595 | 0.595 | 0.027 | −0.033 |
| Vowel | Average | 0.381 | 0.363 | −0.009 | N/A |
| | S.D. | 0.010 | 0.021 | 0.000 | N/A |
| | Min | 0.363 | 0.331 | −0.009 | N/A |
| | Max | 0.392 | 0.388 | −0.009 | N/A |
| Cloud | Average | 0.461 | 0.458 | −0.006 | N/A |
| | S.D. | 0.014 | 0.012 | 0.000 | N/A |
| | Min | 0.449 | 0.450 | −0.006 | N/A |
| | Max | 0.478 | 0.478 | −0.006 | N/A |
| CMC | Average | 0.441 | 0.442 | −0.002 | N/A |
| | S.D. | 0.002 | 0.002 | 0.000 | N/A |
| | Min | 0.439 | 0.439 | −0.002 | N/A |
| | Max | 0.444 | 0.444 | −0.002 | N/A |

As is evident from Table 1, the K-RBBSO algorithm generated a lower silhouette coefficient corresponding to fewer data compared to the other algorithms and generated a

higher silhouette coefficient corresponding to more data types. In contrast, the SA algorithm and ABC algorithm obtained relatively low evaluation values corresponding to a large proportion of the data. Moreover, the ABC algorithm was unable to derive any results corresponding to Vowel, Cloud, and CMC data.

The SA algorithm also generated negative evaluation values corresponding to Vowel, Cloud, and CMC data. The termination condition was removed for the SA and ABC algorithms corresponding to Glass, Vowel, Cloud, and CMC data, and the algorithms were repeated until results comparable to those of RBBSO and K-RBBSO were achieved. However, even after 12 h, the results derived were not appreciably better than those listed in the table.

The execution times of the K-RBBSO, RBBOS, S.A, and ABC algorithms are presented in Table 3. K-RBBSO was fast, corresponding to all data. Because of its specialized termination condition based on its algorithm's characteristics, the SA algorithm was also fast. However, as evidenced in Table 2, it was unable to derive a suitable evaluation value. The ABC algorithm was unable to obtain results corresponding to Vowel, Cloud, and CMC data.

**Table 3.** Results of calculation time.

| Data Name | | K-RBBSO | RBBSO | SA | ABC |
|---|---|---|---|---|---|
| Iris | Average | 2.973 | 2.949 | 12.600 | 3.200 |
| | S.D. | 0.334 | 0.135 | 0.699 | 2.300 |
| | Min | 2.113 | 2.723 | 12.000 | 1.000 |
| | Max | 3.315 | 3.166 | 14.000 | 7.000 |
| Wine | Average | 6.104 | 6.355 | 35.200 | 4.300 |
| | S.D. | 1.701 | 1.940 | 1.317 | 0.483 |
| | Min | 4.376 | 3.895 | 34.000 | 4.000 |
| | Max | 9.090 | 11.156 | 38.000 | 5.000 |
| Glass | Average | 18.755 | 20.193 | 40.500 | 8.200 |
| | S.D. | 6.336 | 8.303 | 1.716 | 1.874 |
| | Min | 9.688 | 9.844 | 38.000 | 5.000 |
| | Max | 28.606 | 35.466 | 43.000 | 11.000 |
| Vowel | Average | 620.514 | 634.797 | 428.900 | N/A |
| | S.D. | 84.979 | 40.503 | 71.086 | N/A |
| | Min | 528.314 | 579.078 | 345.000 | N/A |
| | Max | 770.179 | 714.604 | 508.000 | N/A |
| Cloud | Average | 1592.942 | 1976.802 | 932.600 | N/A |
| | S.D. | 360.610 | 859.481 | 15.771 | N/A |
| | Min | 687.720 | 1422.610 | 902.000 | N/A |
| | Max | 2094.100 | 3775.230 | 951.000 | N/A |
| CMC | Average | 2096.927 | 2932.954 | 1466.800 | N/A |
| | S.D. | 353.531 | 1341.953 | 35.241 | N/A |
| | Min | 1901.350 | 1916.462 | 1419.000 | N/A |
| | Max | 3036.950 | 5234.389 | 1534.000 | N/A |

However, the quantity of data was proportional to the difference between the execution times of K-RBBSO and RBBSO. Corresponding to CMC data, K-RBBSO exhibited an execution time of 2096.927 s, and RBBSO recorded an execution time of 2932.954 s.

## 4. Discussion

Based on the results obtained in this study, Table 4 lists the advantages and disadvantages of each algorithm. K-RBBSO is a good algorithm for big data. The RBBSO algorithm is also effective but has a longer calculation time than K-RBBSO because RBBSO randomly sets the initial solution. Moreover, the K-RBBSO algorithm can overcome the limitations of SA and ABC algorithms; that is, the previous result has a negative effect on the next calculation, and the silhouette cannot be applied in the original search function.

**Table 4.** Advantages and disadvantages of each algorithm.

| Algorithm | Basis | Advantage | Disadvantage |
|---|---|---|---|
| K-RBBSO | K-means for initial solution setting and silhouette coefficient-based BBO algorithm | - Silhouette coefficient-based results<br>- Suitable for big data (more than 1000 of data in this study)<br>- A slightly faster calculation speed | - Unsuitable for big data (about 800 of data under in this study) |
| RBBSO | Silhouette coefficient (not Euclidean distance)-based BBO algorithm | - Silhouette coefficient-based results<br>- Derive higher results than K-RBBOS from fewer data | - Time increasing on by initial solution setting time. |
| S.A. | An algorithm modeled after the quenching process for steel handling | - Searching local solutions with a broad base of evaluation results | - Reforming the solution searching function for using silhouette<br>- Unable to derive accurate results from big data<br>- More detailed parameter setting needs (for example, t, T and K etc.)<br>- Previous results affect the next calculation |
| A.B.C. | An algorithm modeled after the process by which bees collect honey | - Best solution search conditions can be adjusted through parameter setting | - Stopping condition affects results<br>- More detailed parameter setting needs (for example, K, # of pop and bob, etc.)<br>- Previous results affect the next calculation |

The relative efficiency of the BBO algorithm designed by Pal and Saraswat [19] could not be assessed because Euclidean distance is specified as the evaluation function. Thus, it is only applicable to cases with pre-determined numbers of clusters. However, the volume of data is uncertain in arbitrary cases, and the number of clusters must be dynamically decided. In these cases, the silhouette coefficient outperforms the Euclidean distance as the former enables data analysis without prior knowledge of its characteristics.

In this study, the aforementioned fact is verified by evaluating the properties of the silhouette coefficient and by improving the computing speed using immigration, mutation, and emigration habitats, which are core components of the BBO algorithm. In addition, K-RBBSO with k-means is proposed to control the randomness involved in the selection of the initial solution.

Our results indicate that the algorithm proposed in this study is ineffective only corresponding to a limited amount of data. This is corroborated by the calculated silhouette coefficient. However, in the case of the SA or ABC algorithms, the clustering parameters, as well as the data-related factors, require alteration to yield optimal solutions. Moreover, both algorithms obtained low evaluation values corresponding to Glass, Vowel, Cloud, and CMC data, while the ABC algorithm failed to produce any result at all. The SA algorithm includes a self-termination condition, which decreases the termination time progressively at the maximum temperature, T. In this study, the maximum temperature T condition was removed, and the algorithm was allowed to run until results comparable to those obtained using K-RBBSO were obtained. However, an optimum solution could not be obtained even after 12 h, and the ABC algorithm failed to provide any results whatsoever. Moreover, the experiments confirmed the smooth operation of K-RBBSO by merely configuring the number of colonies K, data-related parameters, and the number of habitats without the need to modify any other parameters.

Moreover, the efficient operation of K-RBBSO based on a large number of data was demonstrated. When the volume of data was increased, the SA and ABC algorithms required much longer durations to yield results and identify better solutions. However, K-RBBSO exhibited a lower search time than the regular RBBSO and yielded satisfactory solutions. As presented in Table 3, the average execution time of K-RBBSO over 10 iterations was 2096.927 s, which was nearly 900 s quicker than the 2932.954 s of standard RBBSO. The highest execution time of K-RBBSO was 3036.95 s, compared to 5234.389 s for RBBSO. This is attributed to the poor quality of the initial solution established via random execution in the general RBBSO algorithm. Moreover, the evaluation values obtained using K-RBBSO and conventional RBBSO were not significantly different.

The difference between the silhouette coefficients of K-RBBSO and RBBSO was evaluated (Table 2). The two algorithms exhibited no significant differences corresponding to the Iris and Wine data, but in the cases of Glass, Vowel, Cloud, and CMC data, K-RBBSO yielded somewhat superior evaluation values. This is attributed to the superiority of the determination of the initial solution using K-means—the BBO algorithm relies on the idea that "better solutions produce better habitats"; therefore, the quality of the initial solution is directly correlated to the likelihood of obtaining a better evaluation value.

However, some issues must be considered when using the K-RBBSO algorithm. K-RBBSO is suitable for big data. In this study, the maximum data size was 1473. Therefore, the K-RBBSO algorithm was used.

This study verified the effectiveness of the BBO algorithm. According to Ma et al. [9], the BBO algorithm is the most powerful algorithm among nature-modeled algorithms. Ma et al. conducted a review of the papers related to nature-modeled algorithms such as ABC, BBO, differential evolution (DE), fireflies algorithm (FA), genetic algorithm (GA), group search optimization (GSO), particle swarm optimization (PSO), and shuffled frog leaping algorithm (SFLA). Their study revealed that BBO, DE, FA, and SFLA are the fastest algorithms to solve optimization problems. However, ABC, GA, GSO, and PSO are too slow. The reason for the improved speed of the BBO algorithm is immigration, emigration, and mutation habitat problem-solving. Algorithms using immigration, emigration, and mutation have several problems, one of which is sensitivity. In the previous studies, other optimization algorithms were applied to solve the sensitivity of the BBO formulation problem. Thus, the improved BBO algorithm is more sensitive than the previous BBO algorithm and the other nature-modeled algorithm. The BBO algorithm provides a rationale to reduce the random function dependence of each process so that better results can be gathered. However, other algorithms have not been solved to reduce random function dependence.

## 5. Conclusions

This study focuses on several problems in data clustering—existing formulae require appropriate alterations to make them suitable for the application of the silhouette coefficient; the number of computations required to calculate the silhouette coefficient is very high, and random selection of the initial solution exerts a negative effect on the search for the optimal solution. To overcome these issues, we proposed the K-RBBSO algorithm that uses the silhouette coefficient instead of the Euclidean distance evaluation function, which can only be utilized when the clustering parameters are pre-determined. During the implementation step, the proposed method identifies better solutions based on the silhouette coefficient of each habitat. K-Means is used in this case to reduce the time required to configure the initial solution.

The results of this study can be applied in the following ways. The proposed method can be utilized by users who are contemplating employing clustering to provide personalized services. With the growing ubiquity of the Internet of Things (IoT), it is now possible to safeguard data regarding individuals. Further, a specialized analytical approach for the present generation is necessary for their unique characteristics. However, altering prior classification/evaluation criteria to the present generation is expected to impact the accuracy.

Therefore, clustering is necessary to simultaneously examine large amounts of data [22]. Thus, users who must supply tailored products/services to clients by categorizing them based on customer characteristics can find the conclusions drawn in this study useful. The optimization algorithm follows the principle of the BBO algorithm to "identify better results by gathering good results." As a result, any attempt made to develop an algorithm with the goal of "gathering good solutions to produce better solutions" to tackle local solution issues, such as the BBO algorithm, can be expected to benefit from the conclusions of this study.

Future research should primarily focus on determining the optimal number of habitats. K-RBBSO requires fewer types of parameters to be specified compared to other data clustering algorithms. However, the number of habitats is the most critical parameter—it determines both the execution time and the time required to identify an optimum solution.

Additionally, it is essential to develop a method to reduce the distance calculation process for all data. This is a long-standing problem with the silhouette coefficient. Rousseeuw's [12] silhouette coefficient is effective at identifying more optimum solutions irrespective of the number of clusters. However, because of the evaluation function's characteristics, the distances between all data points must be computed to estimate the relative distance. In future research, a technique to determine relative distances without explicitly measuring the distances between all pairs of data points should be developed.

**Author Contributions:** Conceptualization: S.P.; formal analysis: S.P.; funding acquisition: S.K.; methodology: S.P.; writing—original draft: S.P.; writing—review and editing: S.K. All authors have read and agreed to the published version of the manuscript.

**Funding:** This research was supported by a grant from the National Research Foundation of Korea and funded by the Korean Government (Ministry of Science and ICT; #2020R1A2B501001801). This research received support from the Ministry of Science and ICT, Korea, under the Information Technology Research Center support program (IITP-2021-0-02051) supervised by the Institute for Information & Communications Technology Planning & Evaluation.

**Institutional Review Board Statement:** Not applicable.

**Informed Consent Statement:** Not applicable.

**Data Availability Statement:** Not applicable.

**Conflicts of Interest:** The authors declare no conflict of interest.

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
