# Peer review of "K-RBBSO Algorithm: A Result-Based Stochastic Search Algorithm in Big Data"

_applsci, doi:10.3390/app122312451_

Round 1
Reviewer 1 Report
The article deals with a BBO based methodology to search for an optimum solution stochastically based on the outcomes of a data clustering process. The paper needs improvements and clarifications. I can say that the paper needs a better writing and a better structure. I recommend major revision. The following comments should be considered to improve the paper:
1-The authors have to justify the metaheuristics chosen: Do the BBO algorithm performs well on the problem? If so, the corresponding citation to the specialized literature is required. There are a large number of novel metaheuristic algorithm and several classic algorithm like GA, DE, PSO, CMA-ES ,… the reason behind choosing BBO for the main algorithm and using just SA and ABC algorithms for comparison is not reasonable.
2-The literature survey is really shallow in the area of optimization algorithm. There are only 13 references while there are a wide range of relevant papers published recently. This part must be improved.
3-BBO might be sensitive to the optimization parameters how the parameters were tuned. There should be a table about the parameters of each algorithm (BBO, SA, ABC) and it should be clarified that how they are chosen?
4-How do you ensure that the comparison between the metaheuristics is fair? Each algorithm should be executed for 50 times and then the statistical analysis be presented between the results of different runs. The statistical analysis is not convincing. Please see the following papers for more info about having a fair comparison between metaheuristic algorithms: https://doi.org/10.1007/s10732-008-9080-4, https://doi.org/10.1080/0305215X.2021.1919887.
5-i highly recommend authors to add some other metaheuristic algorithm to their comparison to have a better evaluation of the proposed methodology.
5- there should be a comparison with previous studies. There is no validation and no comparison with previous studies.
6- Conclusion should be concise, please rewrite it with better structure and easy to follow.
Reviewer 2 Report
1. This study proposes an algorithm to search for an optimum solution stochastically based on the outcomes of a data clustering process.
2. The results indicate that K-RBBSO is more effective corresponding to larger volumes of data compared to the other algorithms.
3. This study has creative approach to use K-RBBSO algorithm to overcome shortcoming s of existing stochastic search algorithms.
4. This study creates a data clustering engine and comparing the performance of the K-RBBSO algorithm with those of currently used stochastic search techniques which provide a good tool to resolve the problems of existing stochastic search algorithms.
5. By checking the execution times of the K-RBBSO, RBBOS, S.A, and ABC algorithms in Table 2 & 3, K-RBBSO was verified to be exhibit a lower search time than other algorithms and yielded better and satisfactory solutions to all data.
6. The 9 steps of procedure of K-RBBSO methods are clearly verified in this study.
After reviewing this manuscript, some comments are suggested below:
1. Please clearly describe the background, advantage and shortcoming of the K-RBBSO, RBBOS, S.A, and ABC algorithms.
2. Only 13 references are presented in this manuscript, please add more references within recent 5 years.
3. Please indicate more K-RBBSO applied areas and possible faced problems.
After providing the above minor revision, this manuscript is recommended to be published in the journal of Applied Science.
Round 2
Reviewer 1 Report
Thanks for your efforts. My comments are attached.
